# WARPED CONVOLUTIONAL NEURAL NETWORKS FOR LARGE HOMOGRAPHY TRANSFORMATION WITH $\mathfrak{sl}(3)$ ALGEBRA

## ABSTRACT

Homography has fundamental and elegant relationship with the special linear group and its embedding Lie algebra structure. However, the integration of homography and algebraic expressions in neural networks remains largely unexplored. In this paper, we propose Warped Convolution Neural Networks to effectively learn and represent the homography by $\mathfrak{sl}(3)$ algebra with group convolution. Specifically, six commutative subgroups within the $SL(3)$ group are composed to form a homography. For each subgroup, a warp function is proposed to bridge the Lie algebra structure to its corresponding parameters in homography. By taking advantage of the warped convolution, homography learning is formulated into several simple pseudo-translation regressions. Our proposed method enables to learn features that are invariant to significant homography transformations through exploration along the Lie topology. Moreover, it can be easily plugged into other popular CNN-based methods and empower them with homography representation capability. Through extensive experiments on benchmark datasets such as POT, S-COCO, and MNIST-Proj, we demonstrate the effectiveness of our approach in various tasks like classification, homography estimation, and planar object tracking.

## 1 INTRODUCTION

Convolution Neural Networks (CNN) are famous for their weak translation equivariance in representing visual objects in image space. Essentially, the translation equivariance is achieved due to the constraint and intrinsic topological structure of discrete groups on the image. With a simple group structure, CNN has already been successfully and extensively used in a variety of tasks, including object detection (Dai et al., 2016), recognition (Zhou et al., 2014), tracking (Voigtlaender et al., 2020), and alignment (Ji & Telgarsky, 2020). To further exploit the representation capability, researchers try to extend the conventional convolution to group convolution (MacDonald et al., 2022; Cohen & Welling, 2016; Zhang, 2019; Sosnovik et al., 2020b) with the diversity of group structures.

Among these group structures, the special linear ($SL$) group and its embedding Lie algebra have great potential in visual representation since the corresponding homography describes the relation of two image planes for a 3D planar object with perspective transformation. Every element in $SL(3)$ represents a homography of two different cameras shooting at a static 3D planar object in the scene. The corresponding Lie algebra space $\mathfrak{sl}(3)$ describes the difference in a camera's configuration, which means the local changes in Lie algebra coincide with the movement of viewpoint. Intuitively, a laptop is always a laptop wherever you are looking from. Neural networks built on the space of $\mathfrak{sl}(3)$ could achieve the capability of learning homography, which gives, to some extent, equivariance and invariance to the feature representation for visual objects. This property could make the neural networks robust to occlusions and large view changes, and benefit a number of applications, e.g. homography estimation (Japkowicz et al., 2017), planar object tracking (Zhan et al., 2022), feature representation (Jaderberg et al., 2015).

Currently, few researchers have investigated the relation between homography and the $\mathfrak{sl}(3)$ algebra establishing the connection to the corresponding group. Although some task-oriented works (Esteves et al., 2018; Ye et al., 2021; Finzi et al., 2020; Dehmamy et al., 2021; Benton et al., 2020) show the preliminary results in the application, these existing methods are either only capable of dealing with

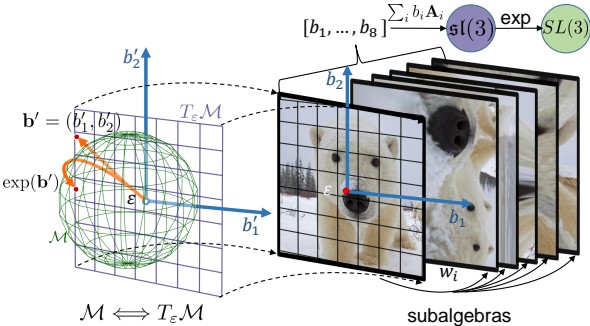

Figure 1: A representation of the relation between a two-parameter commutative Lie subalgebra and the homography. Left $T_\varepsilon \mathcal{M}$ (purple plane) is the tangent space (vector space) of the group's Manifold $\mathcal{M}$ (represented as a green sphere) at the identity $\varepsilon$. We specify it as the two-parameter Lie algebra, where the elements of its generators coefficients vector $\mathbf{b}'$ are orthogonal. With the warp function $w$, we have each subalgebra (right) of the $SL(3)$ satisfying the requirement, the two-parameter transformation thus becomes the translation determined by $\mathbf{b}$, where $\mathbf{b}$ denotes the generators coefficients of $SL(3)$.

several subgroups of $SL(3)$ and their corresponding transformations, or purely enforce the equivariance learning by tremendous data augmentation on the image domain. Subgroup methods (Esteves et al., 2018; Gupta et al., 2021) have the inferior performance in a full perspective transformation setting meanwhile learning equivariance by data augmentation (Ye et al., 2021; Finzi et al., 2020; Benton et al., 2020) or requires multiple samplings MacDonald et al. (2022) have the drawback of heavy computational cost. Our goal is to connect the representation learning of homography with $\mathfrak{sl}(3)$ algebra for neural networks in an efficient way. When the representation is based on Lie algebra, we could investigate the potential in the algebra space with its mathematical property. For instance, the feature representation is consistent with human's intuitive perception, as the transformation walks along the geodesic curve in algebra space as shown in Fig. 1. This allows the networks to have very robust feature representations for different and large transformations and have the capability to neglect the noise in training and improve the data efficiency learning due to the underlying topological structure of the Lie algebra. Additionally, connecting the homography with $\mathfrak{sl}(3)$ algebra is helpful for learning the implicit transformation from a single image, which may facilitate applications such as congealing (Learned-Miller, 2006) and facade segmentation (Xu et al., 2020).

In this paper, we propose Warped Convolution Networks (WCN) to bridge homography to $\mathfrak{sl}(3)$ algebra by group convolution. Inspired by the warped convolution (Henriques & Vedaldi, 2017), we construct six commutative subgroups within the $SL(3)$ group from the Lie algebra $\mathfrak{sl}(3)$ generators to learn homography. For each subgroup, a warp function is proposed to bridge the Lie algebra structure to its corresponding parameters in homography. As the constructed subgroups are Abelian groups, the group convolution operation can be formulated into conventional convolution with a well-designed warp function. By composing these subgroups to form an $SL(3)$ group convolution and predicting several pseudo-translation transformations, our WCN is able to handle non-commutative groups and learn the invariant features for homography in a robust and efficient way. The main contribution can be summarized as follows:

- A framework for efficient Lie group convolution. Our proposed WCN approach can deal with most of the Lie groups by easily combining different basis, which is able to learn the invariant features for homography.
- A robust homography estimator based on WCN. To the best of our knowledge, it is the first work to directly estimate homography along with the $SL(3)$ group and its algebra topology.
- Extensive experimental evaluation on three datasets demonstrates that our approach is effective for various computer vision tasks. It offers the potential for robustly learning the large and implicit transformations.

## 2 RELATED WORK

In this section, we discuss the related prior studies, including equivariant networks and transformation learning. Cohen & Welling (2016) present the fundamental work on equivariance of CNNs representations for the image transformations, where the underlying properties of symmetry groups (Cohen & Welling, 2015) are investigated. They replace the translational convolutions with group convolutions and propose Group equivariant convolutional networks (G-CNNs). For the continuous groups, Cohen & Welling (2016) discretize the group and employ the harmonic function as the irreducible

representation of convolution (Weiler et al., 2018; Zhang, 2019; Sosnovik et al., 2020b). Recently, MacDonald et al. (2022) change the convolution formula and modify the different layers of CNNs. They use the Schur-Poincaré formula for the derivative of exponential map, which enables sampling from Haar measure for arbitrary Lie groups. Dehmamy et al. (2021) propose an unsupervised learning approach to automatically learn the symmetry based on Lie algebra. All these methods introduce the various architectures that are evidently different from the original convolutional layer. It is difficult for them to directly make use of the popular CNN backbones. Moreover, they can only deal with the classification problem.

Early work learns the transformation representation by an auto-encoder (Hinton et al., 2011). It attempts to build a generative model, where the target is a transformed input image. Lin et al. (2021) change the parameterization and project the distance onto $SO(3)$. STN (Jaderberg et al., 2015) introduces a spatial transformation network to manipulate the data in the network without supervision on the transformation. All these methods have difficulty in estimating the transformations, since the networks can only inference once for guessing and the parameters are entangled and highly coupled. ESM (Benhimane & Malis, 2004) parameterizes the arguments as the Lie algebra basis to estimate the $SL(3)$ group. However, their parameters lose the interpretability in an image transformation. Henriques & Vedaldi (2017) employ the warp function on convolution and implement two-parameter group equivariance, since there are possibly utmost two independent dimensions in an image. Recently, deep learning-based approaches predict the homography mainly by estimating the corner offsets (Nguyen et al., 2018; Zhang et al., 2020a) or pixel flows (Zeng et al., 2018). They focus on the local movements in the image space, which are incapable of estimating the large transformation. Our proposed approach can be viewed as a general case of the warped convolution in 2D space, which is able to handle the most sophisticated 2D Lie group $SL(3)$. It is noteworthy that HDN (Zhan et al., 2022) also estimates parameters from two groups based on warp functions (Henriques & Vedaldi, 2017). However, they only employ the rotation-and-scale subgroup and refine the transformation by a corner regression-based estimator, which loses the equivariance for the homography. Differently, our proposed method bridges the gap from the similarity group and two-parameter group to any subgroup of the $SL(3)$ and completes a full homography based on group convolution.

## 3 METHOD

The main objective of this work is to formulate a full homography on Lie subalgebras with several equivariant warped convolutions for 2D projective transformation. Since the warped convolution only implements two-parameter equivariance, a possible way is to combine the several warped convolutions. In general, the 2D projective transformation is an $SL(3, \mathbb{R})$ group having a few subgroups. Our proposed method divides this group into several one or two-parameter subgroups, whose Lie algebras are the subalgebras of $\mathfrak{sl}(3)$. As explained in Fig 1, the warped convolution can be employed to achieve the equivariance for each single or two-parameter transformation. Finally, they are combined to obtain the full transformation. In this section, we first introduce the fundamentals of the warped convolution and then describe our proposed method.

### 3.1 WARPED CONVOLUTION

The key to CNNs' equivariance is their convolution layers, in which the basic operation is the convolution of an image $I \in \mathbb{R}^{n \times n}$ and a convolution kernel $F \in \mathbb{R}^{n' \times n'}$. By employing the Dirac delta function on the image and kernel (Henriques & Vedaldi, 2017), the convolution formula can be treated as a special case of continuous function as follows,

$$(I * F)(\mathbf{v}) = \int I(\mathbf{v} + \mathbf{u})F(-\mathbf{u})d\mathbf{u}. \tag{1}$$

where $\mathbf{u}$ and $\mathbf{v}$ are the coordinates for $I$ and $F$. For the sake of convenience, we shift the image $I$ instead of $F$ in the convolution equations. To prove the equivariance, we define the transformation operator as $\pi_l : \mathbf{u} \longmapsto \mathbf{u} + l$. Hence, the equivariance concerning the translation l can be easily proved as: $(\pi_l(I) * F)(\mathbf{v}) = \int I(\mathbf{u} + (\mathbf{v} + l))F(-\mathbf{u})d\mathbf{u} = (\pi_l(I * F))(\mathbf{v})$.

The standard convolution only takes into account the translation equivariance in the image. For the equivariance of other groups in the image domain, Henriques & Vedaldi (2017) suggest an intuitive solution that defines a function of a group action on the image as below,

$$(\tilde{g} * \tilde{h})(q) = \int_G g(pqx_0)h(p^{-1}x_0)d\xi(p). \tag{2}$$

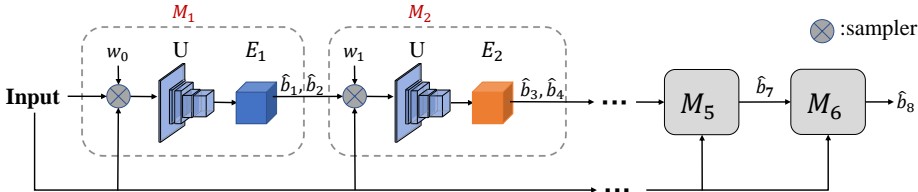

Figure 2: Overview of WCN. $M_i$ is a basic module of WCN, and $w_i$ is the $i_{th}$ warp function. The output of $M_i$ is the estimated Lie algebra coefficients $b_i$. The basic module $M_i$ includes a CNN backbone $U$, and a translation estimator $E_i$. The sampler is used to re-sample the input according to predefined warp function $w_i$ and previously estimated parameters $\hat{b}_i$. We do not specify the detailed structure for its generality, details for specific tasks can be found in the Experiments and Appendix.

Eq. 2 provides the convolution of two real functions $\tilde{g}$ and $\tilde{h}$ in group notation. $g(px_0) = \tilde{g}$ and $h(px_0) = \tilde{h}$ are defined on a subset $\Omega \in \mathbb{R}^2$, where $x_0 \in \Omega$ is an arbitrary constant pivot point. Compared to the convolution on the image, the operation $\mathbf{u} + \mathbf{v}$ becomes $pq$ for the function $g$ and $h$, where $p, q \in G$. The integration is under the Haar measure $\xi$, which is the only measure invariant to the group transformation. Since Eq. 2 is defined over the group, it needs to be further simplified for practical use. As illustrated in Fig 1, a simple approach is to define $G$ as a Lie group, which can be projected onto the Lie algebra $\mathfrak{m}$. $\mathfrak{m}$ is a vector space tangent at identity $\varepsilon$ of the group manifold $\mathcal{M}$, whose base coefficients $\mathbf{b}$ are easy to map to the Cartesian vector space $V \in \mathbb{R}^m$. The dimension $m$ is the degrees of freedom for $\mathcal{M}$. This mapping allows us to estimate the Lie algebra on the real plane $\mathbb{R}^2$. Once the element of Lie algebra is obtained, it could be mapped to $\mathcal{M}$. Therefore, an exponential map $\exp : V \to G$ is employed to connect the Cartesian vector space with the $\mathcal{M}$, where $V$ is a subset of $\mathbb{R}^2$. We therefore have the warped image $g_w(\mathbf{u}) = g(\exp(\mathbf{u})x_0)$. Thus, Eq. 2 can be rewritten as:

$$(\tilde{g} * \tilde{h})(\exp(\mathbf{v})) = \int_V g_w(\mathbf{u} + \mathbf{v})h_w(-\mathbf{u})d\mathbf{u}. \tag{3}$$

where $h_w(\mathbf{u}) = h(\exp(\mathbf{u})x_0)$. Obviously, Eq. 3 has a similar structural form as Eq. 1. This achieves the equivariance to the transformation belonging to the Lie group by performing a conventional convolution after warping the image which connects the warped convolution and group convolution.

## 3.2 WARPED CONVOLUTIONAL NEURAL NETWORKS

As introduced in Section 3.1, warp function is used for implementing the estimation from the Lie algebra, which shares the equivariance and properties of Lie algebra. However, in a warped image $g_w$, one can only estimate at most two independent Lie algebra parameters for its dimensional restriction. To accomplish the goal for estimation purely from the Lie algebra for $\mathfrak{sl}(3)$, we thus employ the compositional method to estimate the Lie subalgebras and combine the subgroups in order. To warp the image by a series of functions, the $SL(3)$ generators need to be defined before the composition. A generator of the Lie algebra is also called the infinitesimal generator, which is an element of the Lie algebra. In this paper, we choose the widely used 2D projective group decomposition (Harltey & Zisserman, 2003), whose corresponding eight generators of its subgroups are defined as follows,

$$\mathbf{A_1} = \begin{bmatrix} 0&0&1 \\ 0&0&0 \\ 0&0&0 \end{bmatrix} \mathbf{A_2} = \begin{bmatrix} 0&0&0 \\ 0&0&1 \\ 0&0&0 \end{bmatrix} \mathbf{A_3} = \begin{bmatrix} 0&-1&0 \\ 1&0&0 \\ 0&0&0 \end{bmatrix} \mathbf{A_4} = \begin{bmatrix} 0&0&0 \\ 0&0&0 \\ 0&0&-1 \end{bmatrix} \mathbf{A_5} = \begin{bmatrix} 1&0&0 \\ 0&-1&0 \\ 0&0&0 \end{bmatrix} \mathbf{A_6} = \begin{bmatrix} 0&1&0 \\ 0&0&0 \\ 0&0&0 \end{bmatrix} \mathbf{A_7} = \begin{bmatrix} 0&0&0 \\ 0&0&0 \\ 1&0&0 \end{bmatrix} \mathbf{A_8} = \begin{bmatrix} 0&0&0 \\ 0&0&0 \\ 0&1&0 \end{bmatrix} \tag{4}$$

For each $i_{th}$ generator $\mathbf{A}_i$, we construct a one-parameter group. The other dimension for $g_w$ could be viewed as an identity transformation group, which is commutative to the one-parameter group. As a result, the equivariance is also valid in the case of one-parameter group. We choose the generators and compose them corresponding to two or one parameter group for warping, as long as they are commutative. In this paper, we propose to compose the generators of $\mathfrak{sl}(3)$ into six Lie subalgebras as $[b_1\mathbf{A_1} + b_2\mathbf{A_2}, b_3\mathbf{A_3} + b_4\mathbf{A_4}, b_5\mathbf{A_5}, b_6\mathbf{A_6}, b_7\mathbf{A_7}, b_8\mathbf{A_8}]$, where $[b_1, b_2, ..., b_8]$ are the elements of the generator coefficients vector $\mathbf{b}$. For better symbol presentation, we re-parameterize $\mathbf{b}$ into a homography-friendly format (Harltey & Zisserman, 2003) to link the Lie algebra with the homography $\mathbf{H}$. The resulting intermediate variables vector $\mathbf{x} = [t_1, t_2, \theta, \gamma, k_1, k_2, \nu_1, \nu_2] = [b_1, b_2, b_3, \exp(b_4), \exp(b_5), b_6, b_7, b_8]$. Therefore, the six Lie subalgebras corresponding subgroups ($\mathbf{H_t}, \mathbf{H_s}, \mathbf{H_{sc}}, \mathbf{H_{sh}}, \mathbf{H_{p1}}, \mathbf{H_{p2}}$) parameterized by $\mathbf{b}$ are defined as follows,

$$\mathbf{H}(\mathbf{x}) = \mathbf{H_t} \cdot \mathbf{H_s} \cdot \mathbf{H_{sc}} \cdot \mathbf{H_{sh}} \cdot \mathbf{H_{p1}} \cdot \mathbf{H_{p2}} \tag{5}$$

$$= \begin{bmatrix} 1&0&b_1 \\ 0&1&b_2 \\ 0&0&1 \end{bmatrix} \begin{bmatrix} \exp(b_4)\cos(b_3)&-\exp(b_4)\sin(b_3)&0 \\ \exp(b_4)\sin(b_3)&\exp(b_4)\cos(b_3)&0 \\ 0&0&1 \end{bmatrix} \begin{bmatrix} \exp(b_5)&0&0 \\ 0&\exp(-b_5)&0 \\ 0&0&1 \end{bmatrix} \begin{bmatrix} 1&b_6&0 \\ 0&1&0 \\ 0&0&1 \end{bmatrix} \begin{bmatrix} 1&0&0 \\ 0&1&0 \\ b_7&0&1 \end{bmatrix} \begin{bmatrix} 1&0&0 \\ 0&1&0 \\ 0&b_8&1 \end{bmatrix}.$$

Based on the above equation, we propose Warped Convolutional Networks (WCN) to learn homography by six modules $[M_1, ..., M_6]$ as depicted in the Fig. 2. Each module $M_i$ has three components, a shared backbone $U$, a translation estimator $E_i$ and a warp function $w_i$. According to Eq. 3, recovering the Lie algebra parameters is equivalent to estimating a pseudo-translation in the algebra space to which the warp function transfers the image space. For each module $M_i$, the input image is resampled with a specially designed warp function $w_i$, fed to the backbone $U$ and estimator $E_i$ to obtain a pseudo-translation in the corresponding subalgebra. Note that we predict $\mathbf{b}$ essentially, and $\mathbf{x}$ is just a function of $\mathbf{b}$ for a convenient expression. $E_i$ is different for each module to adapt to the different subalgebras. Finally, we obtain the output $\mathbf{x}$ and compose them to the transformation matrix as Eq. 4. Please refer to the Appendix for more details.

### 3.3 WARP FUNCTIONS

As illustrated in Eq. 3, the key to recovering one or two-parameter transformation is to find a proper warp function so that the pseudo-translation shift in the warped image is equivalent to the linear changes of element on the corresponding Lie algebra. We thus define the warp function as $w(\mathbf{b}') = \mathbf{u}'$, where $\mathbf{b}' = (b_1', b_2')$ is the specific two-parameter coefficient vector of the warp function for two-parameter Abelian group. For one-parameter group, $b_2'$ is the identity Lie algebra parameter $b_\epsilon$. $\mathbf{u}' = (u_1', u_2')$ denotes the re-sampled point in the transformed image $I$. $\mu = (\mu_1, \mu_2)$ is adopted to denote the point coordinate in the warped image $g_w$.

**Scale and Rotation**  CNNs are equivariant to the translation that is preserved after feature extraction. As a result, $w_0$ is an identical function and we omit it in our implementation. For the scale and rotation group $\mathbf{H}_s$, $\gamma$ represents the uniform scale, and $\theta$ denotes the in-plane rotation. As described in (Henriques & Vedaldi, 2017; Zhan et al., 2022), the warp function $w_1$ for two Lie algebra coefficient parameters $b_3$ and $b_4$ is defined as:

$$w_1(b_3, b_4) = \mathbf{u}'^T = \begin{bmatrix} s^{\gamma'} \cos(b_3) \\ s^{\gamma'} \sin(b_3) \end{bmatrix}. \tag{6}$$

where $s$ determines the degree of scaling and $\gamma = s^{\gamma'}$. We have $\gamma = (s^{\gamma'} = e^{\gamma' \log s}) = e^{b_4}$. Let $s$ be a constant, and estimating $\gamma'$ is equivalent to finding the Lie algebra element $b_4$. Fig. 3 (a) shows the example warp functions for the scale and rotation.

**Aspect Ratio**  Group $\mathbf{H}_{sc}$ represents aspect ratio changes, whose corresponding element of Lie algebra is $b_5$. Since there is a redundant dimension, we employ the warp function with two vertical directions in order to double-check the parameter $b_5$. The corresponding warp function is defined as follows,

$$w_2(b_5, -b_5) = \left[ s^{k_x'}, s^{k_y'} \right]^T. \tag{7}$$

where $k_1 = (s^{k_x'} = \exp(k_x' \log s)) = \exp(b_5)$ and $1/k_1 = (s^{-k_x'} = \exp(-k_x' \log s)) = \exp(-b_5)$. Estimating $k_x'$ and $k_y'$ is actually to find the $b_5$ of $\mathfrak{sl}(3)$. To improve the accuracy, we flip the other quadrant image to the positive quadrant and upsample it to the original image size of $n \times n$. This changes the size of the warped image $g_w^{n \times n \times 4}$. Fig. 3 (b) shows the example result of $g_w$.

**Shear**  Shear transformation, also known as shear mapping, displaces each point in a fixed direction. According to the following equation on point $\mathbf{u}$, it can be found that shear is caused by the translation of each row in the original image, in which the translation degree is uniformly increased along with the column value. $\mathbf{u}^*$ is the transformed points as below,

$$\mathbf{u}^* = \mathbf{H}_{sh} \cdot \mathbf{u}^T = [u_1 + k_2 u_2, u_2]^T. \tag{8}$$

Inspired by the fact that the arc length of each concentric circle with the same angle increases by the radius uniformly, it is intuitive to project the lines onto a circle arc so that the shear can be converted into rotation. Similar to the warping in Eq. 6, the rotation is eventually formulated into the translation estimation. The warp function for shear group $\mathbf{H}_{sh}$ can be derived as follows:

$$w_3(b_6, b_\epsilon) = [b_6 b_\epsilon, b_\epsilon]^T. \tag{9}$$

where $b_\epsilon$ is the unchanged coordinate for one-parameter group. Fig. 3 (c) shows an example of shearing in the horizontal direction.

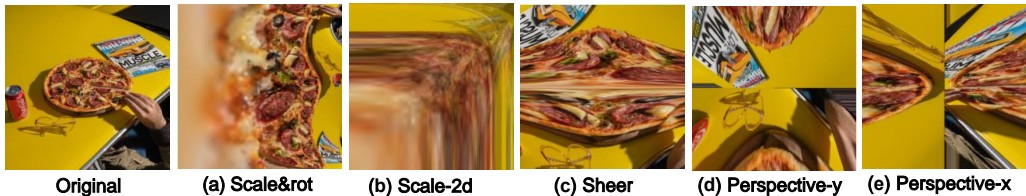

| Original | (a) Scale&rot | (b) Scale-2d | (c) Sheer | (d) Perspective-y | (e) Perspective-x |

Figure 3: The examples after using the warp function for the original image.

**Perspective**   The two elements $\nu_1$ and $\nu_2$ reflect the perspective distortion of an image, which is not the same as the previous one due to view change.

$$\mathbf{u}^* = \mathbf{H}_p \mathbf{u}^T = \mathbf{H}_{p2}\mathbf{H}_{p1}\mathbf{u}^T = \left[\frac{u_1}{\nu_1 u_1 + \nu_2 u_2 + 1}, \ \frac{u_2}{\nu_1 u_1 + \nu_2 u_2 + 1}\right]^T \quad (10)$$

where $\mathbf{H}_p$ denotes the transformation for perspective change. From the action of the group $\mathbf{H}_p$ in Eq. 10, it can be found that the slope of any point does not change after the transformation. $b_7$ and $b_8$ are entangled in Eq. 10. We design two one-parameter warp functions to account for the perspective changes of two groups $\mathbf{H}_{p1}$ and $\mathbf{H}_{p2}$.

$$w_4(b_7, b_\epsilon) = \left[\frac{1}{b_7}, \ \frac{b_\epsilon}{b_7}\right]^T, w_5(b_\epsilon, b_8) = \left[\frac{b_\epsilon}{b_8}, \ \frac{1}{b_8}\right]^T. \quad (11)$$

There exist serious distortions when this sampling function is used as explained in Appendix. The larger the radius is, the more sparse the sampling points are. To tackle this issue, we select the patch near the center of the warped image. Fig. 3 (d,e) shows examples of the perspective warped image in two directions. Please check the Appendix for details of all the warp functions.

## 4   EXPERIMENTS

Our proposed WCN is designed for tasks that are potentially related to the single-plane transformation. To demonstrate the effectiveness of our proposed approach, we evaluate the WCN framework on three visual learning tasks including classification, planar object tracking, and homography estimation. All of them need to recover the underlying homography of the object. To this end, we have conducted experiments on several datasets including MNIST (LeCun & Cortes, 2005), POT (Liang et al., 2018), and Synthetic COCO (S-COCO) (Lin et al., 2014). More details and experiments can be found in the Appendix.

### 4.1   CLASSIFICATION ON MNIST-PROJ

MNIST handwriting dataset (LeCun & Cortes, 2005) is a small testbed for digit classification. We perform the experiment on it to show the effectiveness of WCN on the classification tasks and the general benefit of homography learning in visual tasks. Specifically, we generate the MNIST-Proj dataset by augmenting the data in the training process with projective transformation. The testing dataset has 10,000 digits images and the size of samples is $28 \times 28$.

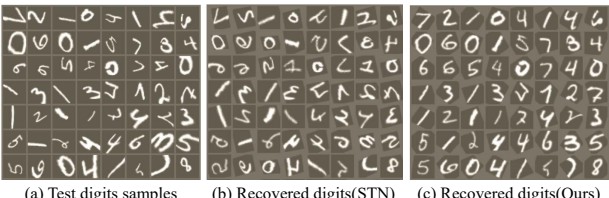

(a) Test digits samples  (b) Recovered digits(STN)  (c) Recovered digits(Ours)

Figure 4: The visual results of the randomly selected MNIST testing after STN and WCN.

Usually, a homography recovery-based method requires a template reference.  For the classification problems, there is no explicit reference object to learn. Inspired by the congealing tasks (Learned-Miller, 2006), we learn an implicit template pose, where the template is the upright digits in MNIST. Please refer to the Appendix B.2 for more implementation details.

Table 1: MNIST-Proj results.

| Methods | Type | Error (%) | Time(ms) |
|---|---|---|---|
| L-conv (Dehmamy et al., 2021) | Any | 19.16 ($\pm$1.84) | 1.81 |
| homConv (MacDonald et al., 2022) | $SL(3)$ | 14.72 ($\pm$0.72) | 105.7 |
| PDO-econv (Shen et al., 2020) | $p(6)$ | 1.66 ($\pm$0.16) | **0.14** |
| LieConv (Finzi et al., 2020)) | Any | 2.7 ($\pm$0.74) | \ |
| PTN (Esteves et al., 2018) | $Sim(2)$ | 2.45($\pm$0.66) | \ |
| STN (Jaderberg et al., 2015) | $Affine$ | 0.79 ($\pm$0.07) | 0.20 |
| **WCN (Ours)** | $SL(3)$ | **0.69** ($\pm$0.09) | 0.42 |

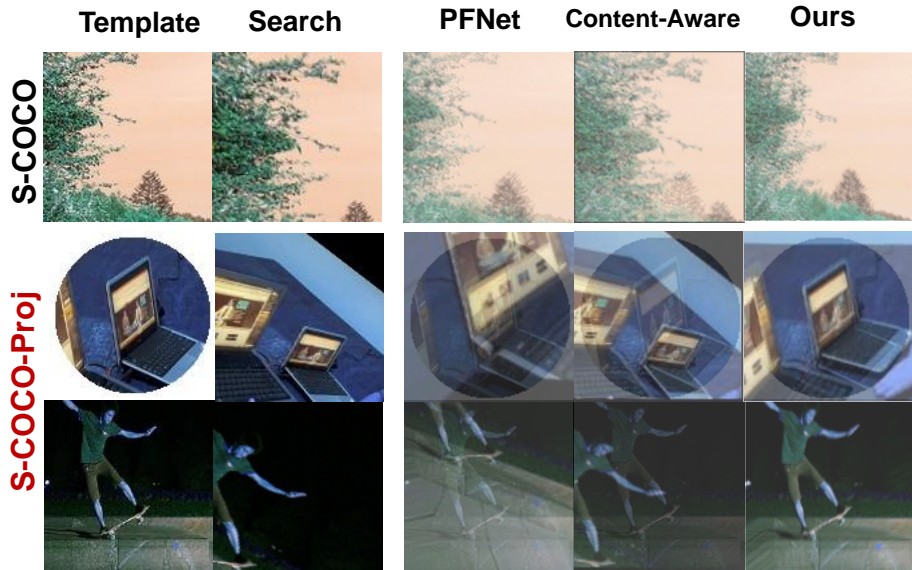

Figure 5: Comparisons of different homography estimation datasets and the mainstreamed methods. We warp the search according to the estimation and merge it with the template. In the second row, we add the occlusion on the corners in the template. The more clear without drifting the result image is, the better performance the corresponding method has.

The error rate is adopted as the metric for evaluation, which is calculated by the total number of wrongly predicted samples divided by the total number of samples. In Table. 1, our proposed framework outperforms six other methods. We use the official implementations for those methods, and all the methods are trained with perspective transform augmentations. Compared with homConv (MacDonald et al., 2022) which is equivariant to the $SL(3)$ group, our method has a significantly improved performance from 0.69 to 14.72 due to the feature level invariant of our proposed work. Although L-conv (Dehmamy et al., 2021) and LieConv (Finzi et al., 2020) are built based on Lie algebra theoretically, WCN still outperforms these two methods due to the robustness of our representation learning in $SL(3)$ group. PDO-econv (Shen et al., 2020) and PTN (Esteves et al., 2018) handle the rotation well, yet we still attain a lower error rate. As shown in Fig. 4, the visual results show the advantage of our proposed WCN over STN in recovering the homography, which can be utilized directly in other tasks like homography estimation. Our proposed WCN achieves more robust and interpretable homography learning compared to other works. Please refer to Appendix.D for more experiments.

Table 2: S-COCO comparison.

| Methods | MACE |
|---|---|
| PFNet (Zeng et al., 2018) | 1.73 |
| PFNet*+biHomE (Koguciuk et al., 2021) | 1.79 |
| HomographyNet (DeTone et al., 2016) | 1.96 |
| UnsupHomoNet (Nguyen et al., 2018)) | 2.07 |
| Content-Aware (Zhang et al., 2020a) | 2.08 |
| PFNet* (Zeng et al., 2018) | 1.20 |
| **WCN (Ours)** | 10.80 |
| **WCN+PFNet*** | **0.73** |

Table 3: S-COCO-Proj comparison.

| Methods | MACE | |
|---|---|---|
| | Mid. Aug. | Lar. Aug. |
| Content-Aware (Zhang et al. (2020a)) | 40.57 | 56.57 |
| HomographyNet (DeTone et al. (2016)) | 19.17 | 35.59 |
| PFNet* (Zeng et al. (2018)) | 11.86 | 25.30 |
| PFNet*+biHomE (Koguciuk et al. (2021)) | 12.62 | 33.12 |
| **WCN (Ours)** | **10.23** | **17.73** |
| PFNet* (w.o. occlusion ) | 2.45 | 13.84 |
| WCN (w.o. occlusion) | 6.29 | 11.31 |
| **WCN+PFNet* (w.o. occlusion )** | **0.69** | **1.81** |

## 4.2 HOMOGRAPHY ESTIMATION

**S-COCO** S-COCO is a commonly used dataset for evaluating the homography estimation task performance. We follow the settings of DeTone et al. (2016) and thereby test our method on S-COCO. Table. 2 shows the experimental results and indicates that our method has more advantages in improving robustness for homography estimation. S-COCO is designed for homography estimator and consists of small transformation as shown in Fig. 5. By adding our proposed WCN to the PFNet*(the current best implementation of PFNet), the standard MACE error (Mean Average Corner

Error) (Zeng et al., 2018) decreases from 1.21 to 0.73, which indicates that our WCN gives an effective homography representation and has generality in improving performance.

**S-COCO-Proj** To better demonstrate the robustness of our WCN, we synthesize a challenging synthetic dataset with large homography changes and occlusion based on COCO14 (Lin et al., 2014) with over 40000 images. Fig. 5 shows the visualization of the difference between S-COCO-Proj and SCOCO datasets, and the qualitative results of different methods. It can be seen that other methods mainly aim at recovering the transformation of the static large scene with minor transformation, while our WCN considers large transformation and the robustness of the corner occlusion.

Table 3 exhibits the performance when using S-COCO-Proj in large transformation, middle transformation, and occlusion settings (see B.4 for more details). We adopt the standard MACE metric to evaluate the performance. Our proposed approach outperforms the other methods, especially with large transformations and occlusions. This demonstrates the robustness of the proposed approach in real-world scenarios due to the homography representation learning in $\mathfrak{sl}(3)$. For non-occlusion settings, our WCN can be used as a robust homography representation for the SOTA method and Ours+PFNet* achieves the highest accuracy performance and is significantly better than using them separately. Similar to S-COCO settings, this indicates consistent results that our homography representation is robust to occlusion and large transformation. Our proposed WCN can be plugged into and boost the performance of those methods that benefit from a decent initialization.

### 4.3 PLANAR OBJECT TRACKING

**POT** POT (Liang et al., 2018) is a challenging mainstreamed planar object tracking dataset that contains 210 videos of 30 planar objects with 107100 images in the natural environment. We select the perspective changes category videos as our testing datasets to evaluate the performance in perspective transformation estimation. The state-of-the-art visual object trackers and planar object tracking approaches (Zhan et al., 2022; Zhang et al., 2020b) show that it is easy to predict the offset of an object through the cross-correlation (Bertinetto et al., 2016). Thus, we employ it as the parameter estimator in our framework. Please refer to Appendix.C for more details.

We choose three methods that directly estimate the Lie algebra coefficient elements for comparison. LDES (Li et al., 2019) takes advantage of the log-polar coordinate to estimate the similarity transform. HDN (Zhan et al., 2022) employs a deep network to estimate the similarity parameters, in which a corner offsets estimator is used to refine the corner. ESM (Benhimane & Malis, 2004) directly estimates 8 Lie algebra elements in image space whose generators are different from ours.

We adopt two metrics for evaluation, including precision and homography success rate as defined in (Liang et al., 2018). The experimental result is shown in Fig. 6. Compared with other trackers, our proposed method achieves a higher average precision and success rate. It has a much

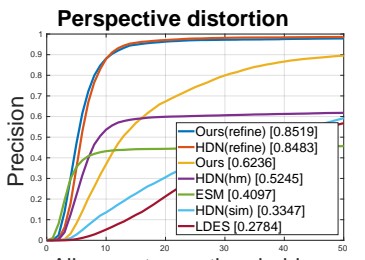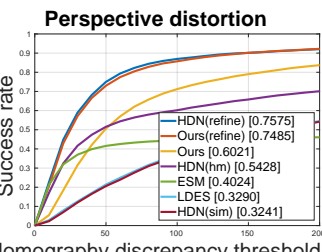

Figure 6: Comparisons on POT. Precision (left), Success rate (right).

higher precision when the error is larger (up to 90%), which indicates the high robustness of our method with 2D perspective transformation. Without the refinement component in HDN (HDN(sim), our model (0.6236) outperforms it (0.3347) by 28.89%. Meanwhile, with the refinement component as same as HDN, our WCN achieves comparable performance for two metrics and higher precision when the error threshold is low.

**POT-L** Similar to S-COCO-Proj, we construct POT-L to better demonstrate the capability of our proposed method. We resample the POT every ten frames to form a new challenging dataset with large transformations. This means the transformation is 10 times larger compared with the original POT. Table. 4 exhibits the results compared to the SOTA method HDN. Our WCN outperforms HDN on the average precision (Avg Prec.) by 3.5%, and WCN performs better for three different error thresholds $e$ consistently. This indicates that our proposed homography representation can be used in various applications and has state-of-the-art performance in planar object tracking task.

Table 4: WCN and HDN comparison on POT-L.

| Tracker | Avg Prec. | Prec.($e \leq 5$) | Prec.($e \leq 10$) | Prec.($e \leq 20$) |
|---|---|---|---|---|
| HDN | 0.685 | 0.091 | 0.488 | 0.805 |
| Ours | 0.720 | 0.123 | 0.521 | 0.833 |

Table 5: Data efficiency analysis.

| Method | 1% | 5% | 20% | 40% |
|---|---|---|---|---|
| STN | 19.99% | 3.16% | 1.45% | 1.14% |
| Ours | 7.16% | 2.19% | 1.27% | 1.01% |

### 4.4 ROBUSTNESS AND DATA-EFFICIENCY

To evaluate the robustness of the proposed method under $\mathfrak{sl}(3)$ algebra, we further test with a wide range of parameters b in a similar setting as in Sec.4.1. As there are two directions for each $i_{th}$ parameter $b_i$, it is hard to analyze them together. We thereby conduct the experiment on each module $M_i$ separately on MNIST-Proj. Each module is trained and tested with a large corresponding parameter range ([L,R]). Fig. 7 shows the result, where L is the left boundary for the transformation parameter, and R denotes the right boundary. We plot examples for every parameter resulting in transformed images. The gray surface marked the standard 95% accuracy level, our WCN achieves a large proportion over this threshold. This confirms a satisfying upper bound for a large transformation range, and our proposed method is able to walk along the $\mathfrak{sl}(3)$ algebra.

Theoretically, our model has equivariance to several groups, thereby it should be easier to train with fewer data. We compare our method against STN on MNIST-Proj with the same setting in Sec.4.1. As shown in Table. 5, our approach has significantly lower error rates with just 1% training data. In our practice, the loss converges much quicker than the other method in the training process.

Table 6: Ablation of backbone on MNIST-Proj.

| Methods | Network | Error (%) |
|---|---|---|
| Naive | LeNet5* | 11.48 ($\pm$1.42) |
| Navie | ResNet18 | 0.87 ($\pm$0.13) |
| STN | LeNet5* | 4.00 ($\pm$0.35) |
| STN | ResNet18 | 0.79($\pm$0.07) |
| Ours | LeNet5* | 3.05 ($\pm$0.33) |
| Ours | ResNet18 | 0.69 ($\pm$0.09) |

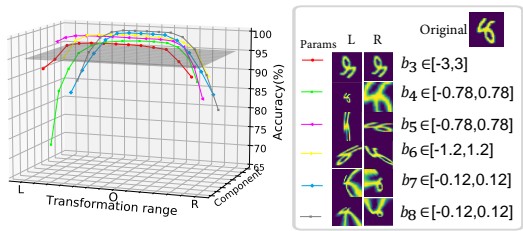

Figure 7: Range robustness for each parameter.

### 4.5 ABLATION STUDY

For fair evaluation, we compare our proposed approach with four baseline methods and compare them with the same backbone on MNIST-Proj. As shown in Table 6, we use the mean error in the last five epochs to measure the performance. When equipped with deeper convolution layers (ResNet18), the CNNs are able to classify the digits well even with large transformations. To fairly compare with STN (Jaderberg et al., 2015), we use the same backbone for classification and achieve a lower error rate. With a five-layer CNN, our proposed approach outperforms STN by 1% and 0.1% with a deeper ResNet18 backbone. This is because the Naive ResNet18 already has the capability of classification and achieves 0.87% compared with STN's 0.79% and Ours 0.69%. Our proposed WCN has consistently better results compared with STN with different backbones. This demonstrates that our proposed method has superior performance of homography learning under $\mathfrak{sl}(3)$ algebra compared with STN.

## 5 CONCLUSION

In this paper, we proposed Warped Convolution Neural Networks (WCN) to effectively learn the homography by $SL(3)$ group and $\mathfrak{sl}(3)$ algebra with group convolution. Based on the warped convolution, our proposed WCN extended the capability of handling non-commutative groups and achieved to some extent equivariance. To this end, six commutative subgroups within the $SL(3)$ group along with their warp functions were composed to form a homography. By warping the corresponding space and coordinates, the group convolution was accomplished in a very efficient way. With $\mathfrak{sl}(3)$ algebra structure, our proposed approach can handle several challenging scenarios, including occlusion and large perspective changes. Extensive experiments showed that our proposed method was effective for homography representation learning and had promising results in three visual tasks, including classification, homography estimation and planar object tracking.

## REPRODUCIBILITY STATEMENT

To ensure the reproducibility, We make efforts on several aspects and conclude here. To keep the generality, we do not specify the concrete structure of WCN. However, we do explain some details of it in Section 4. More details of architecture and implementation for three different tasks can be found in Appendix B. More importantly, we will offer all the source codes for concrete tasks. The training details and experimental setting are introduced in Appendix B.5 and C.1, including how to create the data for our experiment. For reproducibility, we set the random seed for homography estimation, and for MNIST-Proj testing results, we repeatedly test five times and give the error scope.

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

# Appendix

In this appendix, we first discuss the details of the warp function and analyze the influence of each parameter on the warped image. Then, the implementation details of our proposed method are provided, and more experimental details are introduced with additional results. Finally, we provide the proof of warp function property.

## A   WARP FUNCTIONS

For the warped convolution (Henriques & Vedaldi, 2017), the most ideal situation is that the group has commutative property with only two parameters. In this case, all the parameters are independent and the group convolution can be implemented as a warped function. However, it is impossible for both the affine group and projective group to have the same property since they are not Abelian groups with more parameters. A transformation matrix can be represented as follows

$$\mathbf{H} = \exp(\mathbf{A}(\mathbf{b})) = \exp(\sum_{i=1}^{8} b_i \mathbf{A}_i), \tag{12}$$

where $\mathbf{A}_i$ is the generator of the Lie algebra $\mathbf{A}$. $b_i$ is an element of the generator coefficients vector $\mathbf{b}$ in the real plane. For affine and projective group, it does not hold that $e^{\mathbf{A}(\mathbf{b})}e^{\mathbf{A}(\mathbf{a})} = e^{\mathbf{A}(\mathbf{a}+\mathbf{b})}$, where $\mathbf{a}$ is another coefficients vector. Intuitively, this means the coefficients of the SL(3) do not have the additive property, while the proposed corresponding one or two-parameter Lie subalgebras still hold the property. Therefore, no warp function can be found for both affine and projective groups directly satisfying the condition for Eq. (3) in the main paper. It is worthy of discussing why not map the projective transformation onto 3D space to estimate the 6 independent parameters. The reason is that one cannot project the image into a particular camera view without the depth and camera intrinsic. Therefore, there is no way to warp the image like the log-polar coordinates for in-plane rotation (Esteves et al., 2018). In section 3.1 of the main paper, we follow the warped convolution (Henriques & Vedaldi, 2017) and decompose the homography into 6 subgroups that can be predicted independently by pseudo-translation estimation according to the equivariance. Theoretically, the parameters of the proposed six groups may affect each other in the warped image domain. Thus, the groups must be estimated in a cascade fashion. Fortunately, we found that the networks localize the object's position very well in most vision tasks, even with large deformations or distortions. We argue that the networks are capable of learning the invariant feature for the target to compensate for the interdependence. Intuitively, we transfer all 8 parameters of Lie algebra $\mathfrak{sl}(3)$ into 6 subalgebras that can be solved by pseudo-translation estimation. In the warped image domain, the pseudo-translation is more significant in contrast to other transformations. Thereby, we take advantage of this property to estimate the subalgebra in each warped image.

There is little difference in estimating the two-dimension subgroup of $\mathbf{b}$ and predicting the translation. Given the object center as the origin, all transformations generated by the parameters of $\mathfrak{sl}(3)$ do not change the object's center. Unfortunately, this property does not hold for the warped image. The transformation of $\mathbf{b}$ in the warped image is different from the transformation in the original image. To analyze the influence of each parameter on the warp function, we draw the center offset of the warped image. One argument is the parameter $\eta_1$ of the warp function, and the other argument is the other parameter $\eta_2$ may influence the translation. Fig. 8 shows the example for warp function $w_1$. Fig. 8 (a,b,c,d) demonstrate the $k'_x, k_2, \nu_1, \nu_2$ effect on the warped image center offsets compared with $\theta$ about warp function $w_1$. Fig. 8 (e,f,g,h) depict the $k'_x, k_2, \nu_1, \nu_2$ effect on the warped image center in contrast to $\gamma'$ for warp function $w_1$. We find the parameters of $w_1$ dominate the translation of the warped image center. This means that we can estimate $\gamma'$ and $\theta$ in the warped image $g_{w1}$ even with other existing transformations. The same conclusion is valid for other warp functions.

## B   IMPLEMENTATION DETAILS

We design two separate architectures with our proposed WCN for classification, homography estimation and planar object tracking, respectively. In this section, we first give more implementation details of the warp function. Then we introduce the implementation details of the three tasks with datasets MNIST-Proj, S-COCO-Proj and POT accordingly.

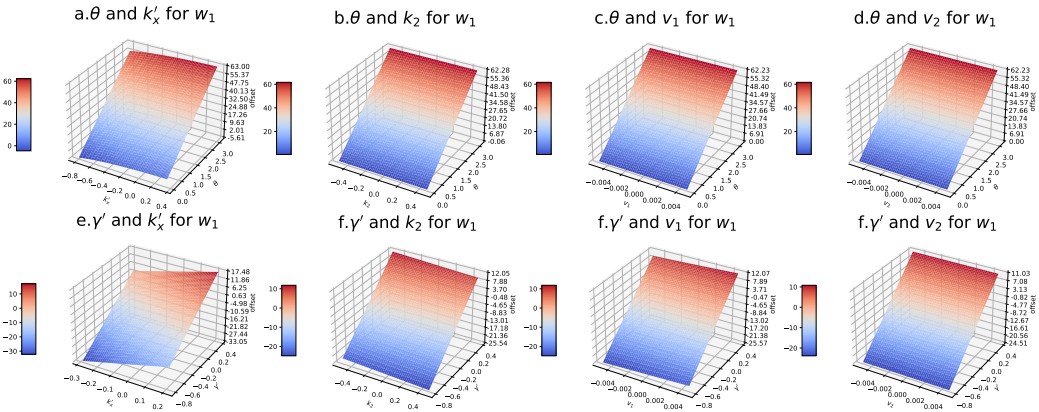

Figure 8: Analysis of the parameter influence to warp function. We use the offsets (pixels) to measure the influence of $x'$ and other parameters.

### B.1 WARP FUNCTIONS

**Scale and Rotation** The range of the parameters should be consistent with the image size by scaling the coordinate in sampling. Therefore, we define the rescaled sampling function for the warped image $g_w^{n \times n}$ according to Eq. (6) in the paper as $(u_1', u_2')^T = [(\frac{n}{2})^{\frac{\mu_1}{n}} \cos(\frac{2\pi\mu_2}{n}), (\frac{n}{2})^{\frac{\mu_1}{n}} \sin(\frac{2\pi\mu_2}{n})]^T$.

Given the warped image with the size of $n \times n$, the warped range is limited by a circle whose radius is $\frac{n}{2}$ in the original image. Let $\hat{\mathbf{b}} = [\hat{b}_1, \hat{b}_2, ..., \hat{b}_8]$ be the prediction of $\mathbf{b}$, $\hat{b}_3$ and $\hat{b}_4$ are recovered by $(\hat{b}_3, \hat{b}_4) = (\frac{2\pi\hat{\mu}_2}{n}, \frac{\hat{\mu}_1}{n} \log(\frac{n}{2}))$, where $(\hat{\mu}_1, \hat{\mu}_2)$ is the prediction of $(\mu_1, \mu_2)$. The mapping function performs on the warped image $\mathcal{W}(I, [\hat{b}_1, \hat{b}_2])$ according to the estimated parameters from $E_1$.

**Aspect Ratio** In the image space, the range of parameters should be consistent with the image size by scaling the coordinate for sampling. The rescaled sampling function for scale estimation in both directions from Eq. (7) in the paper can be derived as $\mathbf{u}'^T = [(\frac{n}{2})^{\frac{2\mu_1}{n}}, (\frac{n}{2})^{\frac{2\mu_2}{n}}]^T$. The corresponding warped image is shown in the Fig. 9 (b). Under the proposed framework, the mapping function performs on the warped image $\mathcal{W}(I, [\hat{b}_1, \hat{b}_2, \hat{b}_3, \hat{b}_4])$ according to the estimated parameters. $b_5$ and $-b_5$ are recovered by $(\hat{b}_5, -\hat{b}_5) = (\frac{2\hat{\mu}_1}{n} \log(\frac{n}{2}), \frac{2\hat{\mu}_2}{n} \log(\frac{n}{2}))$. Since the main task is usually related to an object, its center is treated as the origin of coordinates. In Eq. (7), $w_2 \in (0, +\infty)^2$. Thus, we overpass the other quadrants when $u_1 < 0$ or $u_2 < 0$.

**Shear** In the case of a real image, the rescaled sampling function for shearing from Eq. (9) becomes $\mathbf{u}'^T = \left[\frac{2}{n}(\mu_1 * \mu_2), \quad \mu_2\right]^T$. The estimated $\hat{b}_6$ is recovered by $\hat{k}_2 = \hat{\mu}_1$. Finally, the warp function performs on the warped image $\mathcal{W}(I, [\hat{b}_1, \hat{b}_2, ..., \hat{b}_5])$ according to the estimated parameters.

**Perspective** For transformation $\mathbf{H}_{p1}$, The standard warped image according to paper is depicted in Fig 9(f), which has large distortion and sampling problem. We solve this by setting $w_4 = \left[\frac{\phi_2 n}{2(\mu_1 + sgn(\mu_1)\phi_1)}, \frac{u_2 n}{2(\mu_1 + sgn(\mu_1\phi_1))}\right]^T$, where $sgn$ is the signum function. $\phi_1$ and $\phi_2$ are scaling factors. $w_4$ acts on the warped image $\mathcal{W}(I, [\hat{b}_1, \hat{b}_2, .., \hat{b}_6])$ and $w_5$ acts on the $\mathcal{W}(I, [\hat{b}_1, \hat{b}_2, .., \hat{b}_7])$ according to the estimated parameters. As the same output size is required in sampling, the warp function in Eq. (11) in the paper for sampling can be derived as $\mathbf{u}'^T = \left[\frac{n}{2\mu_1}, \frac{\mu_2 n}{2\mu_1}\right]^T$ and $\mathbf{u}'^T = \left[\frac{\mu_1 n}{2\mu_2}, \frac{n}{2\mu_2}\right]^T$. $\hat{b}_7$ and $\hat{b}_8$ are recovered by $(\hat{b}_7, \hat{b}_8) = (\hat{\mu}_1, \hat{\mu}_2)$.

### B.2 CLASSIFICATION

Two backbone networks are used for the classification task in MNIST-Proj. The first one is a modified LeNet-5 (LeCun et al., 1998). As described in Table. 7, the localization stage is used to predict the pseudo-translation of the handwritten digits on a warped image compared to the implicit upright digits. Then, we resample the image according to the parameters and concatenate it with the original image as the input for the classification stage. To further examine the capability of our method in the

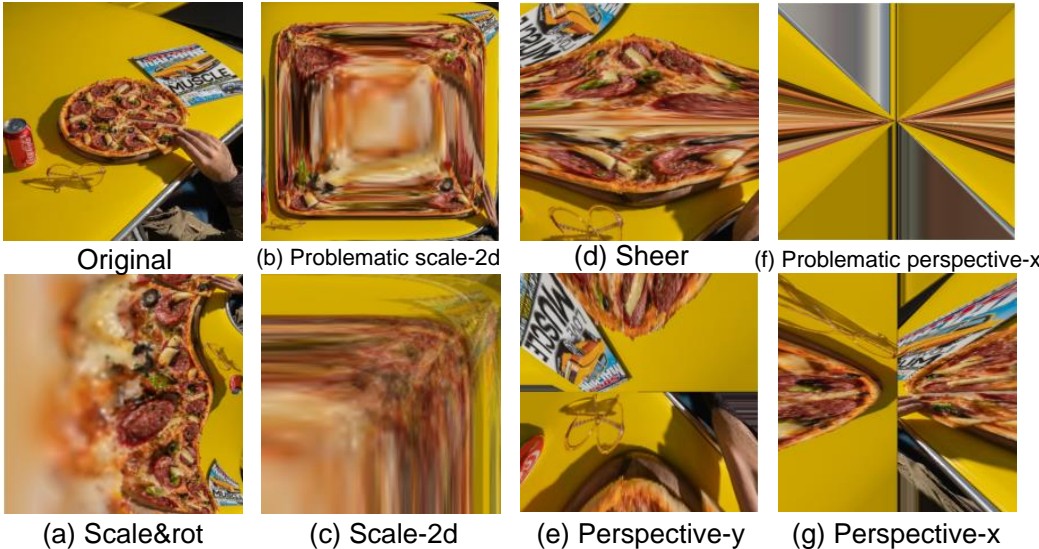

| Original | (b) Problematic scale-2d | (d) Sheer | (f) Problematic perspective-x |
| (a) Scale&rot | (c) Scale-2d | (e) Perspective-y | (g) Perspective-x |

Figure 9: The examples after using the warp function for the original image.

classification task, we implement another classifier to demonstrate the results with a deeper backbone ResNet-18 (He et al., 2016). As listed in Table 8, we first use the ResNet-18 to extract the feature, then estimate the transformation parameters $\mathbf{b}'$ with several warp functions. Its localization network is the same as the Localization stage in Table 7 except that the input size is different. According to the estimated $\hat{\mathbf{b}}'$, the resampled image, and the original image are concatenated as the input of another ResNet-18 that uses a two-layer classifier to predict the class of the digits. As shown in Fig. 10, the

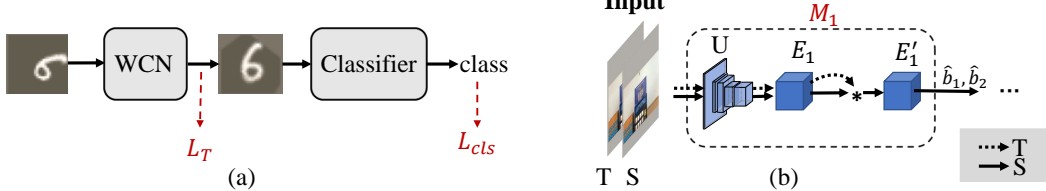

Figure 10: The pipeline of our method for MNIST-Proj digits recognition and planar object tracking tasks. (a): MNIST-Proj digits recognition. $\mathcal{L}_T$: transformation loss, $\mathcal{L}_{cls}$: classification loss. (b): Planar object tracking. This figure only shows changes in a basic component $M_i$, $E_1$ and $E'_1$ are the convolution layers. Here, we use ResNet50 for backbone U. T is the template patch, and the dashed line denotes its data flow. S is the search patch, and the solid line represents its data flow.

pipeline for MNIST classification consists of two components. We first recover the transformation of the digit, and then employ the classifier to predict its class label. We add the supervision both on estimating the transformation parameters using loss function $\mathcal{L}_T$ and image class with loss function $\mathcal{L}_{cls}$. Thus, the total loss is $\mathcal{L}_T + \lambda \mathcal{L}_{cls}$, where $\lambda$ is the weight parameter to trade-off two terms.

## B.3 PLANAR OBJECT TRACKING

For the planar object tracking, we treat HDN (Zhan et al., 2022) as our baseline method, which has two warp functions to predict $\mathbf{b}$. Besides, the perspective changes are small in the feature map. We thereby estimate $p_1$ and $p_2$ directly on the warped image according to the $w_4$ rather than using the correlation. The structure is similar to the homography estimator in HDN, yet we only estimate $\nu_1$ and $\nu_2$ directly.

### B.4 HOMOGRAPHY ESTIMATION

For the homography estimation task, we simply apply the same tracking procedure for estimation. In the testing dataset of SCOCO, transformation is conducted by Eq. (8) in the paper. For middle augmentation, we set $\theta \in [0.6\text{rad}, 0.6\text{rad}]$, $\gamma \in [0.7, 1.3]$, $k_1 \in [-0.2, 0.2]$, $k_2 \in [-0.15, 0.15]$, $\nu_1 \in [-0.0001, 0.0001]$, and $\nu_2 \in [-0.0001, 0.0001]$. Large augmentation is with $\theta \in [0.8\text{rad}, 0.8\text{rad}]$, $\gamma \in [0.7, 1.3]$, $k_1 \in [-0.3, 0.3]$, $k_2 \in [-0.2, 0.2]$, $\nu_1 \in [-0.001, 0.001]$, and $\nu_2 \in [-0.001, 0.001]$. The occlusion on the corner is decided by a circle of radius 60 pixels(image center as the origin), the image over the radius is occluded.

### B.5 TRAINING

Existing datasets lack the annotations of transformation parameters. Even with those provided, they need to be converted to $\mathbf{b}$ with matrix decomposition, which is not easy. Therefore, we augment the possible transformations according to Eq. (8) in the main paper and transform the images as the training data with the randomly sampled parameters. We augment the dataset MNIST (LeCun & Cortes, 2005) for classification and GOT-10K (Huang et al., 2019) and COCO-14 (Lin et al., 2014) for planar object tracking during the training, respectively.

For MINST, the model is trained firstly with the supervision on the predicted $\hat{\mathbf{b}}$ and predicted class for $N_e = 100$ epochs, which is retrained with only classification loss for $N_e$ in MNIST-Proj. For transformation loss, $\mathcal{L}_T = \mathcal{L}_{sr} + \lambda_1(\mathcal{L}_t + \mathcal{L}_{k_1} + \mathcal{L}_{k_2} + \mathcal{L}_\nu)$, where $\mathcal{L}_{sc}$ is the loss of $(b_3, b_4)$. $\mathcal{L}_t$ is the loss of translation, and $\mathcal{L}_{k_1}$ is the loss of $b_5$ and $-b_5$. $\mathcal{L}_{k_2}$ is the loss of $b_6$, and $\mathcal{L}_\nu$ is the loss of $b_7$ and $b_8$. All transformation penalties make use of the robust loss function (i.e., smooth L1) defined in (Girshick, 2015). As for POT, we employ the same classification and offset loss as HDN (Zhan et al., 2022) for the newly added parameters in $\mathbf{b}$.

For MNIST-Proj classification task, we adopt Adam (Kingma & Ba, 2014) as the optimizer, where the batch size is set to 128. The learning rate starts from 0.001 and decays by a multiplicative factor of 0.95 with an exponential learning scheduler. Similar to HDN, we trained the whole network for planar object tracking on GOT-10k (Huang et al., 2019) and COCO14 (Lin et al., 2014) for 30 epochs with 1 epoch warming up. The batch size is set to $28 \times 4$. Our model is trained in an end-to-end manner for 18 hours in our experimental settings.

For S-COCO-Proj homography estimation, we use the same training and testing procedure as in the POT, except that we remove the GOT-10k (Huang et al., 2019) from the training datasets. All the methods in the leaderboard are trained with the same augmented dataset with middle augmentation and mask the corner area with a circle mask with the radius of 60 pixels.

Table 7: Network details for MNIST-Proj. Conv (1,8,7) denotes a convolution layer with input channel=1, output channel=8, kernel size=7. MaxPool (2,2) represents the max-pooling layer with window=2, and stride=2. Linear (90,32) represents the fully connected layer with input size=90 and output size=32. C is the total number of channels for the output.

| Stages | Operator | Output |
|---|---|---|
| | Conv2d (1,8,7) | $C \times 8 \times 22 \times 22$ |
| | MaxPool (2,2), ReLU | $C \times 8 \times 11 \times 11$ |
| Localization | Conv2d (8,10,5) | $C \times 10 \times 7 \times 7$ |
| | MaxPool (2,2), ReLU | $C \times 10 \times 3 \times 3$ |
| | Linear (90,32), ReLU | $C \times 32$ |
| | Linear (32,2) | $C \times 2$ |
| | Conv2d (2,10,5) | $C \times 10 \times 24 \times 24$ |
| | MaxPool (2,2), ReLU | $C \times 10 \times 12 \times 12$ |
| Classification | Conv2d (10,20,5) | $C \times 20 \times 8 \times 8$ |
| | MaxPool (2,2), ReLU, Dropout | $C \times 20 \times 4 \times 4$ |
| | Linear (90,32), ReLU | $C \times 50$ |
| | Linear (50,10) | $C \times 10$ |

Table 8: Classification Networks details using ResNet18

| Stages | Operator | Output |
|--------|----------|--------|
| Classification | Linear (3136,128), Norm,ReLU | C×128 |
| | Linear (128,10), LogSoftmax | C×10 |

Table 9: Ablation on warp functions.

| **Params** | Meaning | **Precision** |
|------------|---------|---------------|
| $[b_1, b_2..., b_8]$ | Perspective | 62.4% |
| $[b_1, b_2..., b_6]$ | Affine | 49.5% |
| $[b_1, b_2..., b_5]$ | Similarity+Shearing | 39.8% |
| $[b_1, b_2..., b_4]$ | Similarity | 33.1% |
| $[b_1, b_2..., b_3]$ | Rotation+Translation | 18.8% |
| $[b_1, b_2]$ | Translation | 13.6% |

## C  EXPERIMENTS

### C.1  EXPERIMENTAL SETUP

We conducted all experiments on a PC with an intel E5-2678-v3 processor (2.5GHz), 32GB RAM, and an Nvidia GTX 2080Ti GPU. The proposed method is implemented in Pytorch.

For MNIST-Proj, the size of the input image is $28 \times 28$. For the hyperparameters of WCN in training, we set $\lambda = 2$, $\lambda_1 = 20$, $\gamma \in [1/1.4, 1.4]$, $\theta \in [-1.5, 1.5]$, $t \in [-28/8, 28/8]$, $k_1 \in [-1.3, 1.3]$, $k_2 \in [-0.03, 0.03]$, and $\nu_1, \nu_2 \in [-0.02, 0.02]$. Due to the numbers 6 and 9 being identical with the rotation even from humans, we remove the number 9 from MNIST-Proj.

For POT, the size of input template $T$ for our networks is $127 \times 127$, while search image $I$ has the size of $255 \times 255$ to deal with the large homography changes. All the hyper-parameters are set empirically, and we do not use any re-initialization and failure detection scheme. For the hyper-parameters of HDN in training, we set $\gamma \in [1/1.38, 1.38]$, $\theta \in [-0.7, 0.7]$, $t \in [-32, 32]$, $k_1 \in [-0.1, 0.1]$, $k_2 \in [-0.015, 0.015]$ and $\nu_1, \nu_2 \in [-0.0015, 0.0015]$.

Many parameters may influence the experimental results. We investigate the influence of the sampling circle radius. We fix it to be $n/2$, which is the half length of the side in the default setting. Theoretically, it is enough as long as the field covers the region of the original image. Actually, we tested 5 different radius $(0.5 \times n/2, 0.75 \times n/2, n/2, 1.25 \times n/2, 1.5 \times n/2)$ The results are quite similar, and the errors are within 0.1%.

### C.2  ABLATION

To evaluate the contribution from each warp function and the adaptability for different groups, we conduct the experiment on POT with different warp functions for transformation parameters. The results are shown in Table 9. It can be seen that the results are better with more warp functions and parameters. Besides, we can combine different warp functions freely.

### C.3  POT

Apart from the perspective distortion in the main paper, we provide more results on other simple transformations, e.g. rotation and scale changes. Fig. 12 shows the precision and success rate of these two transformations. HDN (Zhan et al., 2022) has a similar structure for similarity estimation with two same warp function, which is similar to our presented WCN. Thereby, we have similar results on rotation and scale sequences. Note that, HDN only adopts the previous work (Henriques & Vedaldi, 2017) to their method for similarity transformation. However, our method directly builds on the Lie subalgebras and offers more novel warped functions which have not been proposed before. The proposed method presents a general unified framework to enable the learning the homography. When there are either rotation or scale changes for the object, our WCN estimates all eight parameters rather

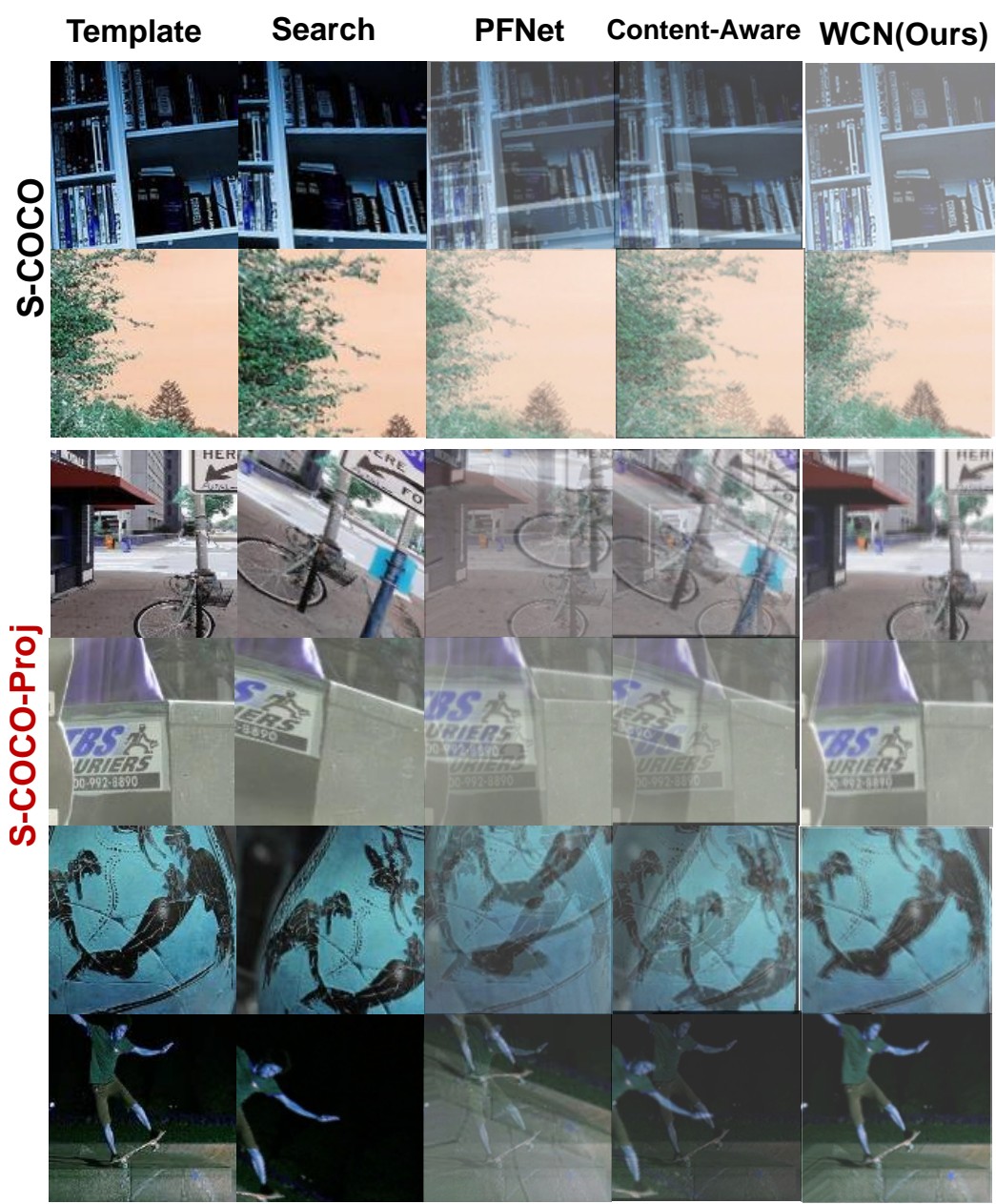

Figure 11: Comparions of different homography estimation datasets and the mainstream methods. We warp the search according to the estimation and merge it with the template. The more clear without drifting the result image is, the better performance the corresponding method has.

than similarity transformation compared to HDN. This may bring more estimation errors because the estimation is not accurate as explained in Sec. A. Furthermore, HDN uses more training data than WCN does.

## C.4 QUALITATIVE RESULTS

**SCOCO & SCOCOProj**

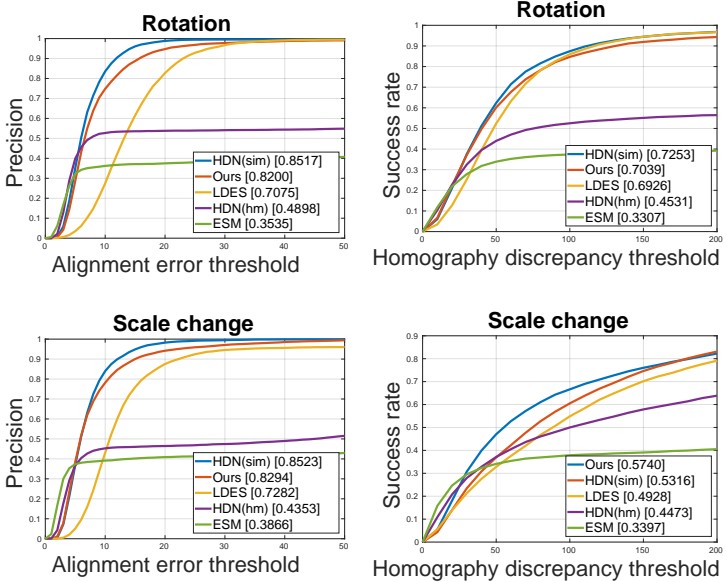

Figure 12: Results on rotation and scale sequences of POT.

We provide more visualization on three different datasets for homography in Fig. 11. With the refined model, the proposed method could robustly estimate the transformation in challenging dataset S-COCO-Proj.

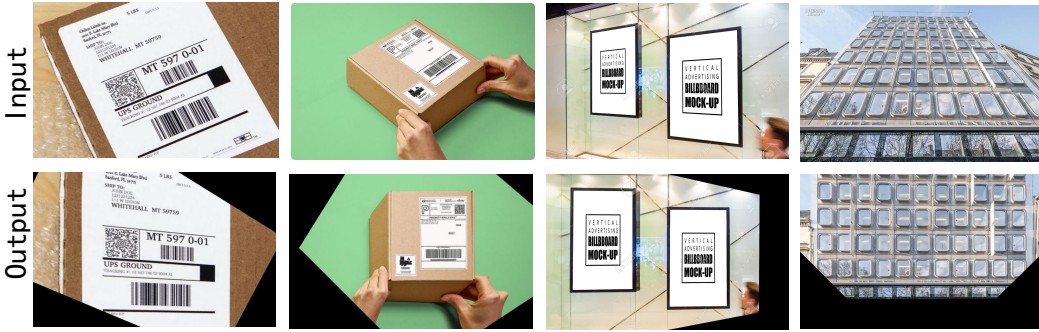

Figure 13: The examples of Planar image dewarping.

**Planar Image Dewarping**

To exhibit the efficiency and performance of the proposed WCN, we apply our searching method based on the Lie algebra elements on the Planar image dewarping. Image dewarping requires only one image, and needs to warp to the original upright view image. This can help in multiple tasks such as building Facade segmentation and document image dewarping. We show some recovered pictures in Fig. 13. The proposed method works well on these images with clear lines for searching the underlying Lie elements. Note that, compared to the other parameterization, the whole search is linear with our method. This greatly reduces the searching complexity from $O(N^8)$ to $O(8N)$.

## D LIMITATION AND FUTURE WORK

Since we build the model on a plane, especially the tasks that require learning the transformation from the template. Thereby tasks like detection only referring to one image are not suitable for the proposed framework. Some methods apply the in-plane equivariant rotation and scale structure (Gupta et al.,

2021; Sosnovik et al., 2020a) to visual tracking or 3D object tracking. They implicitly assume a target is a plane, this does not hold for the more complex 2D projective transformation.

Whereas our proposed method is equivariant to the mentioned several groups, it is hard to theoretically prove that CNNs can robustly estimate a series of transformation parameters sequentially due to the complex multivariate system. Besides, a known problem of WCN is that the estimated offsets may be inaccurate due to the influence of other parameters in addition to $\mathbf{b}'$ and the small feature map size and error produced in different $M_i$. The warped image may thereby involve errors of the previously predicted parameters. To solve this, a larger feature map could be utilized by re-designing the network, and more iterations can be added to refine the error produced in $M_i$. In addition, the proposed method is not robust to the challenging scenarios in our experiments, such as partial occlusions and heavy blur. This could be solved by predicting either an extra occlusion map or a blur kernel in the future work. The proposed method is able to learn the implicit transformation as a result of its special learning space i.e. Lie algebra, meanwhile, it is robust for large transformation and corner occlusion. Therefore, it has great potential for more applications in future work, such as AR, SLAM, recognition, congealing, and image stabilization.

## E  PROOFS

We define the warp function $w$ for different elements in $\mathbf{b}$, and let $\mathbf{b}'$ be the elements in $\mathbf{b}$ with regard to each warp function $w_i$. Although the coordinates of the warped image are proportion to $\mathbf{b}'$, we still need to prove that the group action results on the sampled points $\mathbf{u}'$ in the original image are additive about $\mathbf{b}'$. That is,

$$\mathbf{H}(\Delta\mathbf{b}') \cdot \mathbf{u}'^T \tag{13}$$

$$= \mathbf{H}(\Delta\mathbf{b}') \cdot w(\mathbf{b}') \tag{14}$$

$$= w(\mathbf{b}' + \Delta\mathbf{b}') \tag{15}$$

where $\mathbf{H}$ can be viewed as a function of $\mathbf{x}'$, and $\mathbf{x}'$ can be viewed as a function of $\mathbf{b}'$. $\Delta\mathbf{b}'$ is the incremental value of $\mathbf{b}'$. Therefore, the warped function satisfies Eq. (6) in the main paper, which makes the convolution equivariant to $\mathbf{b}'$.

**Scale and Rotation**
As introduced in the main paper, the warp function for scale and rotation is :

$$w_1(b_3, b_4) = (u_1', u_2')^T = \begin{bmatrix} s^{\gamma'} \cos(b_3) \\ s^{\gamma'} \sin(b_3) \end{bmatrix} \tag{16}$$

We have defined $\gamma = s^{\gamma'} = e^{b_4}$ and $\theta = b_3$, Therefore, the left of Eq. 15 can be rewritten as below:

$$\mathbf{H}_s((\Delta b_3, \Delta b_4)) \cdot \mathbf{u}'^T = \mathbf{H}_s \cdot w_1 \tag{17}$$

$$= \begin{bmatrix} e^{\Delta b_4} \cos(\Delta b_3)u_1' - e^{\Delta b_4} \sin(\Delta b_3)u_2' \\ e^{\Delta b_4} \sin(\Delta b_3)u_1' + e^{\Delta b_4} \cos(\Delta b_3)u_2' \end{bmatrix} \tag{18}$$

$$= \begin{bmatrix} e^{b_4 + \Delta b_4} \cos(b_3 + \Delta b_3) \\ e^{b_4 + \Delta b_4} \sin(b_3 + \Delta b_3) \end{bmatrix} \tag{19}$$

As a result, the warp function $w_1$ supports the equivariance.

**Aspect Ratio**
The warp function for aspect ratio is defined as follows:

$$w_2(b_5, -b_5) = \begin{bmatrix} s^{k_x'}, s^{k_y'} \end{bmatrix}^T, \tag{20}$$

For $\mathbf{H}_{sc}$, there is only one element. We thereby let $k_1 = (s^{k_x'} = \exp(k_x' \log s)) = \exp(b_5)$ and $1/k_1 = (s^{-k_x'} = \exp(-k_x' \log s)) = \exp(-b_5)$. As a result, the left of Eq. 15 can be rewritten as below

$$\mathbf{H}_{sc}((\Delta b_5, -\Delta b_5)) \cdot \mathbf{u}'^T = \mathbf{H}_{sc} \cdot w_2 \tag{21}$$

$$= \begin{bmatrix} e^{\Delta b_5} \cdot u_1' \\ e^{-\Delta b_5} \cdot u_2' \end{bmatrix} = \begin{bmatrix} e^{\Delta b_5 + b_5} \\ e^{-(\Delta b_5 + b_5)} \end{bmatrix} \tag{22}$$

Hence, we can prove the equivariance holds with the warp function $w_2$.

**Shear**

The warp function for Shear is defined as below:

$$w_3(b_6, b_\epsilon) = [b_6 b_\epsilon, \quad b_\epsilon]^T \tag{23}$$

The left of Eq. 15 can be rewritten as:

$$\mathbf{H}_{sh}((\Delta b_6, b_\epsilon)) \cdot \mathbf{u}'^T = \mathbf{H}_{sh} \cdot w_3 \tag{24}$$

$$= \begin{bmatrix} u'_1 + \Delta b_6 u'_2 \\ u'_2 \end{bmatrix} = \begin{bmatrix} (b_6 + \Delta b_6) b_\epsilon \\ b_\epsilon \end{bmatrix} \tag{25}$$

Hence, the equivariance is tenable for $\mathbf{H}_{sh}$ with warp function $w_3$.

**Perspective**

The warp function for perspective can be derived as below:

$$w_4(b_7, b_\epsilon) = \left[ \frac{1}{b_7}, \quad \frac{b_\epsilon}{b_7} \right]^T, w_5(b_\epsilon, b_8) = \left[ \frac{b_\epsilon}{b_8}, \quad \frac{1}{b_8} \right]^T \tag{26}$$

We have defined the $\nu_1 = b_7$ and $\nu_2 = b_8$. Thus, the left of Eq. 15 with regard to $\nu_1$ or $\nu_2$ is rewritten as follows:

$$\mathbf{H}_{p1}((\Delta b_7, b_\epsilon)) \cdot \mathbf{u}'^T = \mathbf{H}_{p1} \cdot w_4 \tag{27}$$

$$= \begin{bmatrix} \frac{u'_1}{u'_1 \Delta b_7 + 1} \\ \frac{u'_2}{u'_1 \Delta b_7 + 1} \end{bmatrix} = \begin{bmatrix} \frac{1}{\Delta b_7 + b_7} \\ \frac{b_\epsilon}{\Delta b_7 + b_7} \end{bmatrix} \tag{28}$$

$$\mathbf{H}_{p2}((b_\epsilon, \Delta b_8)) \cdot \mathbf{u}'^T = \mathbf{H}_{p2} \cdot w_5 \tag{29}$$

$$= \begin{bmatrix} \frac{u'_1}{u'_2 \Delta b_8 + 1} \\ \frac{u'_2}{u'_2 \Delta b_8 + 1} \end{bmatrix} = \begin{bmatrix} \frac{b_\epsilon}{\Delta b_8 + b_8} \\ \frac{1}{\Delta b_8 + b_8} \end{bmatrix} \tag{30}$$

Hence, the equivariance holds for two perspective groups with warp function $w_4$ and $w_5$, respectively.

