# OpenReview forum: "Warped Convolutional Neural Networks For Large Homography Transformation with $\mathfrak{sl}(3)$ Algebra"
_ICLR.cc/2024/Conference — Submitted to ICLR 2024_

### Official Review · Reviewer_o1vN · 2023-10-28

**Soundness:** 3 good
**Presentation:** 3 good
**Contribution:** 3 good
**Rating:** 6
**Confidence:** 4

**Summary:**

This paper proposes Warped Convolutional Networks (WCN) to learn and represent homography by sl(3) algebra with group convolution. The homography is decomposed into six commutative subgroups within SL(3), and the corresponding warp functions are designed to effectively recover the one or two-parameter groups. The experiments on classification, homography estimation, and planar object tracking tasks have shown the superiority of the proposed method.

**Strengths:**

++ This paper shows a good connection between warped convolutions and the SL(3) induced homography.

++ WCN demonstrates better robustness and data efficiency against prior works through comprehensive experiments.

**Weaknesses:**

-- page 7: "improved performance from 0.69 to 14.72"

-- The MACE of PFNet* in Table 2 is 1.20, while it is 1.21 in the main text of page 8.

**Questions:**

1) Does the order of the decomposed subgroups affect the results?

2) Can the estimators be learned in parallel instead of sequentially? The current implementation seems that later estimators need to be conditioned on earlier estimators.

---

> ### Author Response · Authors · 2023-11-19
>
> Thank you for your comments and we appreciate your helpful questions. We highlight your comments and respond to them point by point as follows.
>
> **Q1:"page 7: "improved performance from 0.69 to 14.72."**
>
> Thanks for your careful proofreading. It should be from 14.72 to 0.69. We will revise the manuscript accordingly.
>
> **Q2:"The MACE of PFNet\* in Table 2 is 1.20, while it is 1.21 in the main text of page 8."**
>
> Thanks again and sorry for the typos. After proofreading and double checking, it should be 1.21. We will revise the manuscript accordingly.
>
> **Q3:Does the order of the decomposed subgroups affect the results?**
>
> Yes, and this is an interesting question we haven't investigated thoroughly.
> Our initial finding is that translation must be resolved at first since other Lie parameters need to have a pivot before warping.
> We use the current permutation because of the prior classical decomposition in [1].
> The translation and rotation normally are larger than other parameters, so we put them in the first rank.
> We will investigate this in future work.
>
>
> **Q4:Can the estimators be learned in parallel instead of sequentially? The current implementation seems that later estimators need to be conditioned on earlier estimators.**
>
> We cannot learn them in parallel currently. Theoretically, the Lie algebra SL(3) is not commutative, which means their Lie elements are not purely independent. This is a very tough problem behind the phenomenon. We argue that this is one of the main reasons why it is hard to estimate the homography parameters.
>
> However, it is a very meaningful suggestion and we have considered it before. This will certainly make the model efficient. One possible way now we think is to parallely estimate in a coarse-to-refine fashion.
>
> **References**:
> [1]Multiple view geometry in computer vision, 2003

---

> > ### Comment · Reviewer_o1vN · 2023-11-20
> > **Response to the authors' comments**
> >
> > Thanks for the authors' clarification. I have no further concerns.

---

> > > ### Author Response · Authors · 2023-11-20
> > > **Thanks for your response**
> > >
> > > Dear Reviewer o1vN,
> > >
> > > Thanks very much for your confirmation and your time!

---

### Official Review · Reviewer_i6Q7 · 2023-10-30

**Soundness:** 3 good
**Presentation:** 2 fair
**Contribution:** 2 fair
**Rating:** 5
**Confidence:** 4

**Summary:**

This paper introduces warped convolutional neural network to deal with the task of large homography transformation estimation under SL(3) algebra. 6 commutative subgroups within the SL(3) group are composed to form a homography. For each subgroup, a warp function is proposed, bridging the Lie algebra
structure to its corresponding parameters in homography. Experiments are conducted on the tasks such as classification, homography estimation and planar object tracking.

**Strengths:**

The proposed Lie representation for homography estimation is new and interesting.

**Weaknesses:**

The current presentation is not clearly show its advantages over previous homo representations. Please see questions

**Questions:**

“homography learning is formulated into several simple pseudo-translation regressions”The 4pt representation (corner offsets) of homography estimation is already a 4 translation vector representation, where 4 translational vectors at the 4 corner of an image can uniquely defines a homography. A homography matrix is obtained by solving a DLT

According to Eq. 5, b_i should be estimated in order to estimate a homography. However, these b_i is not with the same value range, for example, the translation b1 and b2 may orders larger than perspective components, b7 and b8, which introduces training difficulty.  This is why previous works adopt corner offsets that share a similar value range for the homography estimation instead of regressing homography matrix elements.  I'm not sure working in the Lie space could solve such problem.

Assume it can, why it is better than corner offsets or homography flow representation is still not clear. Saying that "incapable of estimating the large transformation" is not accurate, given that some works already adopted corner offsets or homography flows for large homography transformations, e.g.,

Jiang et al. Semi-supervised Deep Large-baseline Homography Estimation with Progressive Equivalence Constraint, AAAI 2023

Nie et al. Depth-Aware Multi-Grid Deep Homography Estimation with Contextual Correlation, TCSVT 2021

For experiments, more recent homography dataset should be adopted, e.g., the CA-Homo dataset, or AAAI 23 dataset, should be compared with, in which various deep-based, traditional feature-point matching, deep feature matching based approaches are reported.

Authors should put some efforts to demonstrate the limitations of previous homography representations, so as to show the advantages of the proposed new representation. These previous representations, either corner offsets, or homography flow, on the one hand, can achieve large baseline registration already, on the other hand, are flexible to be extended for multiple planes, e.g., mesh local homos, that can go beyond the single-plane transformation. The flexibility is also important.

---

> ### Author Response · Authors · 2023-11-19
> **To Reviewer i6Q7 (1)**
>
> Thank you so much for the thoughtful comments on the underlying motivation and evaluation of our method.
> We have tried our best to clarify each issue and will sincerely increase more introduction about your concerns in the revision. We are available for more questions. Please find our itemized responses below.
>
> **Q1:...According to Eq.5, $b_i$ should be estimated in order to estimate a homography. However, these $b_i $is not with the same value range, for example, the translation $b_1$ and $b_2$ may orders larger than perspective components, $b_7$ and $b_8$, which introduces training difficulty... I'm not sure working in the Lie space could solve such problem....**
>
>
> Yes, working in the Lie space can deal with difficulty in training. As shown in Table.2 and Table.3, our proposed method exactly solves the regressing homography in Lie space, which demonstrates the promising performance compared to those estimating corner offsets works. This is one of the main contributions of this work.
>
> These $b_i $ are not in the same value range. To alleviate this problem, we convert/warp the original image into a warped space, as shown in the Appendix.A and Fig.8 in the main paper.
> In the warped space/Lie algebra space, the value ranges of these $b_i$ are normalized and can be estimated in a pseudo-translation way, which is another key contribution of our work and why they are easy to predict in this way.
>
>
>
> **Q2:Assume it can, why it is better than corner offsets or homography flow representation is still not clear,..., Authors should put some efforts to demonstrate the limitations of previous homography representations, so as to show the advantages of the proposed new representation. These previous representations, either corner offsets, or homography flow, on the one hand, can achieve large baseline registration already.**
>
> Thanks for your insightful comment. This is an important and valuable question.
> The key issue of estimation on corner translational offsets and recovery by DLT is occlusion in real-world scenarios.
> When you occlude the corners area, the performance of this genre of approaches will decrease hugely (see Table. 3 in the main paper).
> This means the offset of corners (correct points) might not always be the offsets of estimation (wrong points) in the image. Theoretically, if their output regression prediction of the corners is from the other points (if corners are occluded) in the image, it is impossible to learn the underlying homography.
>
> Besides, both the regression of corners [1,2] and flow estimation for the sparse grid in the images [3,4] are local methods due to the convolution mechanism.
> This means that those methods only succeed in small motion/transformation.
> CNN is not equivariant to the translation in large transformations [5], which means the feature is not the same for regression. This makes it hard to train the model.
>
> In our test, the mesh flow-based methods work perfectly on the local and small motion, however, their regression range is usually limited [6].
> To tackle this issue, they usually employ the coarse to refine structure [7], which is complex and engineered. With these refined models, our method could achieve the same accuracy. Moreover, the key points-based methods (deep and traditional) cannot handle the large transformation due to low-texture images [8], and this severely affects their robustness (see Q3 results).
>
> In contrast, our method builds on the Lie algebra, which makes the homography prediction become pseudo-translation regressions.
> Concretely, we can use translation in the Lie algebra space to estimate all parameters of homography, regardless of the transformation in the original image space.
> This makes the prediction problem much simpler and more stable, as it is like estimating the pure translation of the whole image.
> Moreover, our method is more robust to occlusion, as it does not rely on specific features or points in the image.
> We show in Table.3 and Table.5 in the original paper that our method performs better than previous methods on occluded images, as well as on large and diverse transformations.
>
>
> **Q3:For experiments, more recent homography dataset should be adopted, e.g., the CA-Homo dataset, or AAAI 23 dataset, should be compared with, in which various deep-based, traditional feature-point matching, deep feature matching based approaches are reported.**
>
> We thank you for this suggestion. We understand that these datasets are more recent and challenging, and they contain various scenarios and factors that affect the homography estimation. However, we did not adopt these datasets for the following reasons:

---

> ### Author Response · Authors · 2023-11-19
> **To Reviewer i6Q7 (2)**
>
> - We focus more on the theoretical validation rather than the high performance with lots of tricks. We show the effectiveness of our proposed method, which is based on the Lie algebra and the warp function. We keep the basic structure compared to the baseline and do not use tricks to enhance the performance. In our experiment, we already list both the best corner regression and flow-based methods that we could conduct the experiments.
>
> - We think that it would be inappropriate to add more datasets in a single task application, which will confuse readers with what the main contribution of our work is. Our work is a general framework bridging homography to Lie algebra instead of a homography estimator. Three different task experiments including image classification, homography estimation, and planar object tracking have been conducted to demonstrate the effectiveness of our proposed method. Besides, our experiment already shows that we can achieve a similar result after adopting the refinement module compared with existing SOTA methods in S-SCOCO-Proj and SCOCO homography estimation benchmarks. We include more the state-of-the-art results in the following Table.
>
> - Instead of small motion, we address a different problem of large motion and occlusion. Most of the datasets, such as the CA-Homo dataset and AAAI-23 dataset, focus on real scenes with challenges such as low texture, large foreground, low illumination, etc.
> They do not consider the large transformation, which is the main challenge and contribution of our work.
> The transformation range, taking the pixel displacement as a measure, is less than 1/10 of the image size in these datasets.
> In contrast, the displacement can exceed half of the image size (0.43 on average) in our proposed dataset S-COCO-Proj. That’s why we use this new dataset to evaluate our method.
>
> **For methods comparison:**
>
> We appreciate your suggestion to compare with more methods.
> More state-of-the-art methods are added in comparisons as you mentioned (AAAI-23 [7] the resources provided are not enough to reproduce.).
>
> As shown in the table below, we try our best to include a feature matching-based method LoFTR+MagSAC [9,10], a flow-based method MGDH (TCSVT 21 as you mentioned) [4], and an iterative corner regression-based method IHN [2].
> LoFTR+MagSAC performs well in middle transformation. While with large transformation, its performance is inferior to ours.
> Besides, in our testing, their method is not robust for some low texture and large transformation which could not be corrected.
> IHN performs the best with middle transformation with six times iterative refinements.
> However, our model achieves the best with only one-time refinement by PFNet.  All these compared methods could not handle well with large transformations and with occlusion on the corners.
> More results will be included in the revision.
>
> We would like to point out that our paper is mainly about learning the homography structure directly, rather than adopting the consensus techniques to recover the homography. Therefore, we mainly compare with the direct methods that learn the homography parameters from the image features in the main paper.
>
> **Table. 1 Comparisons on S-COCO-Proj**
> | Methods | MACE | |
> |:-----|:-----|:-----|
> |  | **Middle Transformation**| **Large Transformation**|
> | Content-Aware | 40.57 | 56.57 |
> | MGDH | 31.87 | 50.61 |
> | LoFTR+MagSAC | 8.49 | 53.38 |
> | HomographyNet | 19.17 | 35.59 |
> | PFNet   | 11.86  | 25.30|
> | PFNet+biHomeE| 12.62 | 33.12 |
> | IHN      | **5.94**  | 32.95 |
> | **Ours**  | 10.23  | **17.73** |
> | |||
> | **Methods (Without Occlusion on Corners)**|||
> | MGDH  | 19.51 | 40.72 |
> | LoFTR+MagSAC  | 2.37 | 24.07 |
> | PFNet     | 2.45  | 13.84 |
> | IHN     | 0.98  | 13.60 |
> | Ours     | 6.29  | 11.31 |
> | **Ours+PFNet**  | **0.69**  | **1.81** |
>
>
> **Q4:Flexibility for learning multiple plane homography**
>
> Although flexibility for multiple planes may have exceeded our topic, this is an important feature. Our structure is theoretically suitable with multiple planes, a simple approach is to divide it into small patches. We will make an effort in this direction in the future.
>
> **References**:
>
> [1] Unsupervised Deep Homography: A Fast and Robust Homography Estimation Model, RAL 2018
> [2] Iterative Deep Homography Estimation, CVPR 2022
> [3] Rethinking Planar Homography Estimation Using Perspective Fields, ACCV 2018
> [4] Depth-Aware Multi-Grid Deep Homography Estimation With Contextual Correlation TCSVT 2021
> [5] Why do deep convolutional networks generalize so poorly to small image transformations?   JMLR 2019
> [6] Traditional and modern strategies for optical flow: an investigation, SNAS 2021
> [7] Semi-supervised Deep Large-baseline Homography Estimation with Progressive Equivalence Constraint, AAAI 2023
> [8] HDN, AAAI 2022
> [9] LoFTR: Detector-Free Local Feature Matching with Transformers, CVPR 2021
> [10] MAGSAC: Marginalizing Sample Consensus, CVPR 2019

---

### Official Review · Reviewer_uatX · 2023-11-01

**Soundness:** 3 good
**Presentation:** 2 fair
**Contribution:** 3 good
**Rating:** 8
**Confidence:** 4

**Summary:**

This paper introduces a novel approach, named Warped Convolutional Neural Networks (WCN), for effectively learning and representing homography in neural networks through algebraic expressions. The proposed method enables the learning of features that remain invariant to significant homography transformations and can be easily incorporated into popular CNN-based methods. The paper thoroughly analyzes the proposed approach, including the warp function and its properties, implementation details, as well as extensive experimental results on benchmark datasets and tasks. The contributions of this paper encompass a fresh perspective on homography learning utilizing algebraic expressions, the introduction of a novel warped convolutional layer, and a comprehensive evaluation of the proposed method across various benchmark datasets and tasks.

**Strengths:**

1. This paper establishes a sophisticated and elegant relationship between homography and the SL(3) group along with its Lie algebra.
2. The formulation of the homography and the underlying warping functions proposed in this paper demonstrate technical soundness.
3. The proposed WCN method for estimating homography parameters is logical and well-founded.
4. Extensive experiments are performed on various tasks and datasets, successfully validating the effectiveness of the proposed method.

**Weaknesses:**

1. In terms of novelty, this paper bears resemblance to the work of Zhan et al. (2022). Zhan et al. employed a similar approach, utilizing two groups for estimation, whereas this paper proposes the use of six groups for the same purpose. As a result, the contribution of this work can be seen as somewhat incremental. More clear discussion should be given.

2. It is desirable for this paper to provide additional elaboration on the mathematical aspects associated with the proposed method, with the aim of enhancing comprehension for individuals who are not familiar with this particular field. The inclusion of more accessible explanations and intuitive examples would be beneficial in ensuring that the content is more easily understood by a broader audience.

**Questions:**

Please check the weakness listed above.

---

> ### Author Response · Authors · 2023-11-19
>
> We thank you for the valuable comments and constructive suggestions that can greatly enhance the quality of our manuscript. We use bold font to highlight your comments and respond to them point by point as follows.
>
> **Q1:....this paper bears resemblance to the work of Zhan et al. (2022)... More clear discussion should be given.**
>
> Thanks for your suggestion. The main differences between HDN [1] and our proposed method are summarized as follows:
>
> - We aim at building a more general and unified network for the SL(3) group, purely on the algebra space. Meanwhile, HDN presents a method in decomposition to different groups and purely adopts the warp convolution proposed by Henriques et al. [2].
>
> - Thereby, the prediction of our work is the unified variable for the whole warped image. We construct the homography from atomized elements on Lie algebra. However, HDN has to decompose it into two groups and predict the four corners offsets, which introduces the problems of this genre.
>
> - To the best of our knowledge, the additional warp function has not been proposed before. There are lots of excellent research works
>  [3,4,5] that focus on only one-parameter equivariant architectures, HDN instead does not create new warp functions.
> We believe this work is helpful for the community in exploring the related theory on algebraic expression for geometric transformation.
>
> From the perspective of the application, we proposed a new homography estimation dataset S-COCO-Proj.
> None of the existing datasets emphasize the large transformation in homography estimation.
> Most of the existing datasets only have local perspective distortion that is not consistent with the daily observations.
> We start with the Lie algebra, which is able to deal with large transformations in the real-world applications.
>
>
>
> **Q2:....It is desirable for this paper to provide additional elaboration on the mathematical aspects associated with the proposed method**
>
> We thank you for this helpful suggestion. To help the readers better understand our method, we will provide additional information.
>
> - We will include more references to the relevant literature, i.e., group theory and Lie algebra [6,7], for those who want to learn more about the background and the motivation of our work.
>
>
> - We will improve explanations and illustrative examples in the revised manuscript. We understand that the spatial projection and the correspondence are complex and non-linear, and they may not be easy to understand intuitively.
>
> Due to the space limitation, we could not include more details and examples in the main paper.
> We realize that our paper would benefit from more elaboration on the mathematical aspects of our method, as they are essential to understanding the underlying theory and the implementation of our approach.
> We have provided some basic concepts and proofs in Appendix A and Appendix E, which can serve as supplementary materials for interested readers.
>
> **References**:
> [1]Homography decomposition networks for planar object tracking, AAAI 2022
> [2]Warped convolutions: Efficient invariance to spatial transformations, ICML 2017
> [3]Scale-Equivariant Steerable Networks, ICLR 2020
> [4]Rotation equivariant vector field networks, ICCV 2017
> [5]Harmonic networks: Deep translation and rotation equivariance, CVPR2017
> [6]A micro Lie theory for state estimation in robotics, Arxiv 2018
> [7]Lie Groups, Lie Algebras, and Representations: An Elementary Introduction, 2004

---

### Author Response · Authors · 2023-11-19
**Author Response**

**General Response**

We are very grateful to all the reviewers for their valuable feedback and insightful comments on our manuscript.
We will revise our manuscript according to these constructive questions and suggestions that significantly strengthen our paper.

We appreciate that all reviewers acknowledge the novelties and contributions of our proposed method. Especially, reviewer *uatX* and *o1vN* confirm our contribution on bridging algebraic expressions to homography transformation and the effectiveness through comprehensive experiments.  Other questions and suggestions are on different aspects. In the following, we thereby address the specific concerns point by point.

---

### Meta-Review · Area_Chair_t5z3 · 2023-12-15

**Metareview:**

After the rebuttal, one reviewer still has negative comments, and the major issues are: (1) Compared to previous methods, the novelty is not clear, such as Zhan et al. (2022). The authors should demonstrate the limitations of previous homography representations, so as to show the advantages of the proposed new representation.  (2) To evaluate the proposed model, it is better to compare with some SOTA methods on recent homography datasets. After reading the comments, the AC cannot recommend to accept this paper and encourage the authors to take the comments into consideration for their future submission.

**Justification For Why Not Higher Score:**

Please see the detailed comments.

**Justification For Why Not Lower Score:**

Please see the detailed comments.

---

### Decision · Program_Chairs · 2024-01-16

Reject